# Treatment Response Predictors of Neoadjuvant Therapy for Locally Advanced Gastric Cancer: Current Status and Future Perspectives

**DOI:** 10.3390/biomedicines10071614

**Published:** 2022-07-06

**Authors:** Yasushi Sato, Koichi Okamoto, Tomoyuki Kawaguchi, Fumika Nakamura, Hiroshi Miyamoto, Tetsuji Takayama

**Affiliations:** 1Department of Community Medicine for Gastroenterology and Oncology, Tokushima University Graduate School of Biomedical Sciences, Tokushima 770-8503, Japan; 2Department of Gastroenterology and Oncology, Tokushima University Graduate School of Biomedical Sciences, Tokushima 770-8503, Japan; okamoto.koichi@tokushima-u.ac.jp (K.O.); kawaguchi.tomoyuki@tokushima-u.ac.jp (T.K.); nakamura.fumika.2@tokushima-u.ac.jp (F.N.); miyamoto.hiroshi@tokushima-u.ac.jp (H.M.); takayama@tokushima-u.ac.jp (T.T.)

**Keywords:** gastric cancer, neoadjuvant chemotherapy, biomarker, metabolic enzymes, nucleotide excision repair, liquid biopsy, miRNA

## Abstract

Neoadjuvant chemotherapy (NAC) for locally advanced gastric cancer (LAGC) has been recognized as an effective therapeutic option because it is expected to improve the curative resection rate by reducing the tumor size and preventing recurrence of micrometastases. However, for patients resistant to NAC, not only will operation timing be delayed, but they will also suffer from side effects. Thus, it is crucial to develop a comprehensive strategy and select patients sensitive to NAC. However, the therapeutic effect of NAC is unpredictable due to tumor heterogeneity and a lack of predictive biomarkers for guiding the choice of optimal preoperative treatment in clinical practice. This article summarizes the related research progress on predictive biomarkers of NAC for gastric cancer. Among the many investigated biomarkers, metabolic enzymes for cytotoxic agents, nucleotide excision repair, and microsatellite instability, have shown promising results and should be assessed in prospective clinical trials. Noninvasive liquid biopsy detection, including miRNA and exosome detection, is also a promising strategy.

## 1. Introduction

Gastric cancer is the fifth most common cancer worldwide and the third leading cause of cancer-related deaths. The highest mortality rates from gastric cancer have been reported in East Asia, while the lowest rates have been reported in North America [1].

Curative resection with no residual cancer, both macroscopically and histologically, is the only way to provide a cure for gastric cancer. However, in many locally advanced gastric cancers (LAGC), curative resection may not be possible due to invasion of the surrounding organs or advanced lymph node metastasis. Even in cases where curative resection is achieved, most patients experience locoregional and/or distant relapse, and the long-term survival rate remains unsatisfactory. The high risk of post-surgery recurrence has led to the development of relapse-preventing strategies to improve survival. This has led to the investigation of adjuvant therapy or neoadjuvant approaches, including chemotherapy and chemoradiotherapy. There is accumulating evidence for a variety of adjuvant therapy options to improve survival, such as adjuvant systemic chemotherapy, typically used in Asian countries; perioperative chemotherapy (neoadjuvant plus adjuvant therapy), mainly used in European countries; and adjuvant chemoradiation, historically preferred in North America. However, a consensus on the best treatment options from Western and Eastern countries is yet to be determined due to the heterogeneous nature of the disease [2,3].

Neoadjuvant chemotherapy (NAC) chemotherapy has several benefits and risks compared to postoperative chemotherapy (Table 1) [4,5,6]. As such, the clinical benefits of NAC remain controversial, especially in Japan [6]. In fact, a pivotal randomized phase III trial, the Japan Clinical Oncology Group (JCOG) 0501, compared the survival benefit of S-1 plus cisplatin (CDDP), as NAC in 300 patients with resectable type 4 or large type 3 gastric cancer with that of surgery and S-1 as adjuvant chemotherapy [7,8]. The 3-year overall survival (OS) was 62.4% (95% confidence interval, 54.1–69.6) in the control group and 60.9% (52.7 to 68.2) in the neoadjuvant group with a hazard ratio of 0.916 (0.679–1.236; *p* = 0.284), suggesting that NAC with S-1 plus cisplatin failed to demonstrate a survival benefit. 

In Europe, the Medical Research Council Adjuvant Gastric Infusional Chemotherapy (MAGIC) trial compared pre-and postoperative chemotherapy (epirubicin, CDDP, and 5-fluorouracil [5-FU]; [ECF]) with surgery alone [9]; it was observed that perioperative (neoadjuvant) chemotherapy had superior OS rates. Therefore, it has become the mainstay for treating LAGC. Recently, the taxane-containing FLOT regimen (docetaxel, oxaliplatin, leucovorin, and 5-FU) showed superiority over ECF in terms of histological response, relapse-free survival, and OS [10,11]. The greatest benefit from perioperative chemotherapy appears to come from preoperative NAC because even in the Arbeitsgemeinschaft Internistische Onkologie (AIO)-FLOT4 trial, less than half of the study population completed the postoperative treatment as outlined in the protocol. Currently, there are no approved targeted or immune checkpoint inhibitors in the perioperative setting; however, there are many ongoing trials designed to examine the efficacy of these agents in various combinations [12].

It is crucial to understand which patients will benefit from NAC because predicting the histopathological response to NAC can significantly affect patient outcomes [13]. However, the optimal approach for each patient is still not straightforward and remains controversial, which can be partly explained by the lack of predictive tools for perioperative treatment in routine clinical practice [14]. Therefore, it is necessary to identify patients who will benefit from NAC, and the ability to predict chemosensitivity from NAC should be an area of intense investigation, especially in the age of precision medicine. Ideally, the predictive value of a biomarker to a specific NAC should be determined from material obtained before the treatment by using endoscopic specimens or blood of the patients because post-treatment samples may not accurately reflect the original biology of the tumor due to the impact of the treatment itself.

In this manuscript, we provide an overview of the current status of predictive biomarkers for a histopathological response to NAC in LAGC and discuss the limitations and future perspectives. This includes tissue- or blood-based biomarkers for NAC, as well as predictors of response to therapy using liquid biopsy with micro RNAs (miRNAs) and exosomes, which are expected to be developed in the future. 

## 2. Biomarkers Involved in NAC

To date, 5-FU/CDDP-based combination therapy has been widely used for NAC [15], but due to drug resistance, the single-drug efficacy rates are not more than 20%, and the overall efficacy rate of first-line chemotherapy based on 5-FU or CDDP is less than 50%. Thus, some patients cannot benefit from NAC [16].

Therefore, there is an urgent need to explore the indicators of enzyme profiles related to CDDP and 5-FU or S-1 metabolism as predictors of the response to treatment. The most examined indicators include thymidylate synthase (TS), thymidine phosphorylase (TP), dihydropyrimidine dehydrogenase (DPD), and excision repair cross-complementation group 1 (ERCC1). In addition, apoptosis-associated proteins, histone demethylases, microsatellite instability (MSI), miRNAs, and exosomes have been reported as potential predictors (Table 2).

## 3. Metabolic Enzymes Associated with 5-FU Resistance

In the body, 5-FU is phosphorylated by TP to form the metabolically active substance fluorodeoxyuridine monophosphate and binds to TS, which is necessary for DNA synthesis, and forms a ternary complex with reduced folate. This ternary complex inhibits DNA synthesis, resulting in the suppression of cell proliferation. However, more than 85% of 5-FU is reduced to inactive metabolites by DPD in the liver and other organs and excreted through the kidneys. Therefore, the activity of DPD plays an important role in the efficiency of 5-FU [26].

In 2002, Terashima et al. [27] reported that the activity of DPD in gastric cancer tissues could predict drug resistance to 5-FU. Later, Napieralski et al. found that patients with a high expression of DPD were not sensitive to 5-FU and had a poor prognosis, whereas the opposite was observed in patients with a low expression of DPD [17]. In addition, Wang et al. detected TS overexpression in a DNA microarray analysis of 5-FU-resistant cancer cell lines [28]. A meta-analysis of 555 gastric cancer patients treated with S-1 showed a significant difference in response rate depending on the expression of DPD. However, there was no significant difference in the overall response rate based on the expression levels of TS and TP [29]. In contrast, Ott et al. identified a TS tandem repeat polymorphism in blood samples as an independent prognostic factor in the NAC group in LAGC patients treated with 5-FU-based preoperative chemotherapy [18]. A significant improvement in survival was also observed in the 2rpt/2rpt and 2rpt/3rpt genotypes [18]. These results suggest that TS and DPD are useful markers among the enzymes related to fluorouracil metabolism, but their clinical significance in NAC has not been fully established and further studies are needed.

## 4. Nucleotide Excision Repair (NER)

The NER pathway repairs relatively widespread DNA damage—several tens of base pairs—caused mainly by UV light [30,31]. The major NER pathways and other functional protein complexes are responsible for complicated NER reactions [32]. After DNA damage, ERCC1 forms a complex with XPA, XPF, and RPA proteins and binds to the DNA damage site for subsequent cleavage, removal, and repair of the damaged DNA. Platinum drugs, such as CDDP, induce cell death by forming cross-links within and between DNA strands. Since cross-linked adducts are suitable substrates for NER, the relationship between ERCC1 expression and CDDP sensitivity has attracted much attention. Metzger et al. reported that the expression of ERCC1 mRNA correlated with prognosis in 38 patients with gastric cancer treated with preoperative chemotherapy (CDDP+5FU) [19]. Similar results were also reported by Wei et al. in a study using a modified folinic acid, fluorouracil, and oxaliplatin (FOLFOX) regimen [33]. 

In addition, Fareed et al. reported that CDDP-based preoperative chemotherapy for gastroesophageal adenocarcinoma had significantly better pathologic tumor reduction and survival in ERCC1-negative tumors diagnosed using immunohistochemistry, which can be used as a predictive marker for treatment [20]. Kwon et al. reported that the response and survival rates to chemotherapy were significantly better in patients with ERCC1-negative tumors than in those with ERCC1-positive tumors in a study involving 64 patients treated with 5-FU and oxaliplatin before surgery. Moreover, ERCC1 expression was a prognostic factor in a multivariate analysis [34]. We investigated the relationship between the expression of major NER proteins and treatment responses in patients enrolled in a phase II study of docetaxel, CDDP, and S-1 (DCS) NAC for LAGC and found that damaged DNA binding protein complex subunit 2 (DDB2) and ERCC1 were associated with the treatment response [21,35]. DDB2 is known to be a sensor protein for early damage recognition during NER, and loss of its function increases the susceptibility of cancer cells to DNA damage [36]. To investigate the relationship between the expression of ERCC1 and/or DDB2 and the clinical effect of DCS therapy, we examined the expression of these proteins in tumor tissues before treatment by immunohistochemistry and analyzed the correlation with the anti-tumor effect (pathological response) of DCS therapy. The results showed that the positive predictive rates of ERCC1 and DDB2 expression for predicting resistance to DCS therapy were 72.9% and 78.3%, respectively. The positive predictive rate for predicting resistance to DCS therapy was as high as 82.5% when both were combined, suggesting their potential to be useful as markers of resistance to preoperative DCS therapy. 

## 5. Apoptosis-Related Molecules

The relationship between apoptosis-related molecules and gastric cancer chemotherapy resistance has also been investigated. For example, low expression of BAX has been associated with lower response rates in patients treated with 5-FU in combination with CDDP [37], capecitabine, oxaliplatin plus irinotecan (COI), or FOLFOX [38]. BCL2-homology domain 3 (BH3) proteins, such as BAD, BIM, and BID, activate BAX and inhibit anti-apoptotic factors of the intrinsic apoptotic pathway. Altering the expression of these proteins may promote chemoresistance in gastric cancer. Therefore, we investigated gastric cancer cell lines by using a BH3 profiling method [39,40], which quantitatively evaluates the dependency of apoptosis on BH3 peptides, and found that docetaxel-induced apoptosis correlates with BIM and BAK protein expression, and that BAK knockdown causes docetaxel resistance [22]. We also determined the BAK expression index of 69 gastric cancer specimens before DCS therapy by multiplying the BAK positivity with a number representing the staining intensity. We found that patients with a good histopathological response to chemotherapy had a higher BAK expression index than those with incomplete response, and those with a BAK expression index of three or higher had a better progression-free survival and overall survival, indicating that BAK protein expression can predict the antitumor effect of docetaxel-containing therapy using pretreatment biopsy tissues. Similarly, Wu et al. also reported that decreased expression of BIM was associated with decreased overall survival in docetaxel-treated patients [41].

## 6. Histone Demethylation

Histone methylation can positively or negatively affect gene transcription, and dysregulation of histone methyltransferases is known to be involved in tumorigenesis [42]. Among them, Jumonji domain-containing protein 2A (JMJD2A), a member of the JMJD2 family, catalyzes the demethylation of H3K36 or H1.4K26. JMJD2A is overexpressed in a variety of cancers and promotes tumor growth [43], and is associated with drug resistance and poor clinical outcomes [44,45]. Using microarray analysis of gene expression in pretreatment biopsies of gastric cancer, Nakagawa et al. identified a functional gene signature consisting of 29 genes that are predictive of response to DCS therapy [46], among which JMJD2A was involved in gastric cancer chemosensitivity. They showed that overexpression of JMJD2A was positively correlated with the response rate in 34 patients treated with DCS [47]. These findings suggest that histone demethylation may be a novel epigenetic factor that regulates sensitivity to chemotherapy for gastric cancer.

## 7. Microsatellite Instability (MSI) and Epstein-Barr Virus (EBV)

MSI expression has been reported as a predictor of chemotherapy efficacy and response to immune checkpoint inhibitors [48]. However, the benefit of perioperative chemotherapy in MSI-high (MSH-H) gastric cancer remains controversial, due to the limited number of these patients in various clinical studies [49].

For example, in a meta-analysis of postoperative adjuvant chemotherapy for resectable gastric cancer, MSI-high (HIS-H) status was shown to be a negative predictor of prognostic benefit from adjuvant chemotherapy. In a meta-analysis of individual patient data from MAGIC, the Capecitabine and Oxaliplatin Adjuvant Study in Stomach Cancer (CLASSIC), the Adjuvant Chemoradiotherapy in Stomach Tumors (ARTIST), and the Intergroup Trial in Adjuvant Chemotherapy for Adenocarcinoma of the Stomach (ITACA-S) trials, Pietrantonio et al. found that the 5-year overall survival rate in the MSI-H group was significantly prolonged than in the MSI-low and microsatellite stable (MSS) groups (77.5% vs. 59.3%). Furthermore, they reported that additional chemotherapy was effective in the MSI-low/MSS group but not in the MSI-H group (70% vs. 77%) [50]. Hashimoto et al. investigated the expression of MLH1 and PD-L1 in surgical specimens from 110 and 285 patients who were treated with NAC and surgery alone, respectively. The results showed that the response rate to preoperative chemotherapy was significantly lower in MLH1-negative patients than in MLH1-positive patients, but there was no significant difference between patients with high and low PD-L1 expression. Conversely, the relapse-free survival of patients who did not receive preoperative chemotherapy was significantly longer in the MLH1-negative group than in the MLH1-positive group, and there was no significant difference in relapse-free survival between the two groups in patients who received preoperative chemotherapy. In addition, PD-L1 expression was not associated with relapse-free survival in patients with or without chemotherapy [23]. Similarly, Haag et al. reported a poor histological response to NAC in MSI-H tumors [24]. Therefore, it is suggested that MLH1-negative or MSI-H gastric cancers are unlikely to have a histological response to NAC. It is expected that immune checkpoint inhibitors can be used against them in the future. 

Interestingly, Biesma et al. reported substantial histopathologic responses after NAC in patients with MSI-high gastric cancer, but only those with a mucinous phenotype from the D1/D2 trial in which patients underwent surgery alone and the ChemoRadiotherapy after Induction chemoTherapy In Cancer of the Stomach (CRITICS) trial in which patients underwent surgery and perioperative treatment [51,52]. These results indicate that the mucinous phenotype may be a relevant parameter in future clinical trials for patients with MSI-H.

Epstein–Barr virus-positive (EBV+) gastric cancer is one of the distinct molecular subtypes in The Cancer Genome Atlas (TCGA) classification [53]. Patients with EBV-negative gastric cancer have better outcomes than patients with EBV-negative and microsatellite stable (EBV−/MSS) gastric cancer [50,53,54,55,56]. However, data on response rates to NAC in EBV+ resectable gastric cancer are limited [57]. In the CRITICS trial, among the molecular subgroups of gastric cancer, EBV+ tumors had the highest histopathologic response rate and better outcomes than EBV-/MSS tumors [49].

In one retrospective series, Kohlruss et al. reviewed 760 NAC cases and found that MSI-H and EBV+ do not predict response to platinum- and 5-FU-based NAC but indicates a good prognosis. Particularly, MSI-H indicates a good prognosis regardless of treatment with NAC. Since MSS predicts a good response to NAC and suggests a poor prognosis for patients treated with surgery alone, MSS may help identify patients who would benefit more from preoperative chemotherapy [54].

## 8. miRNA and Exosomes

Micro RNAs are single-stranded small RNAs with a length of approximately 18–25 nucleotides, of which more than 1000 have been identified [58]. miRNAs regulate gene expression by binding to the 3′ untranslated region of mRNA and are involved in regulating a variety of biological processes [59]. 

Recently, miRNAs have been investigated as possible molecular markers and are expected to be used for the diagnosis [60,61] and prognosis of various cancers [62], as well as for predicting the effects of anticancer drugs. For example, let-7i is an miRNA involved in chemoresistance. Liu et al. examined the tissues of 86 patients with LAGC who had undergone preoperative chemotherapy and curative resection. They found that a lower level of let-7i expression in tumor tissues prior to treatment was associated with a lower histological response rate to NAC (FOLFOX regimen), indicating that let-7i expression could be a predictive marker of chemotherapy resistance in patients with LAGC [25]. Tan et al. found that the expression levels of miR-145 and miR-185 in the peripheral blood of 120 patients undergoing NAC with S-1 and oxaliplatin (SOX) tended to be lower in the tumor progression group, indicating that miR-145 and miR-185 may help predict the efficacy of SOX therapy when used as NAC [63]. 

miRNAs are present in cell-secreted vesicles called exosomes, which protect miRNA from degradation in the bloodstream and allow for the detection of miRNA in the blood [64,65]. Therefore, liquid biopsy, which is a method to detect exosomal miRNAs secreted by tumor cells, has been attracting considerable attention as a promising method for monitoring chemoresistance [66,67,68]. For example, Zhang et al. reported that CDDP and paclitaxel promoted the secretion of miR-522 in exosomes from cancer-associated fibroblasts, suppressed arachidonate lipoxygenase 15 (ALOX15), and decreased the accumulation of lipid-ROS (toxic lipid peroxides) in gastric cancer cells, resulting in reduced sensitivity to anticancer drugs [69]. In addition, Wang et al. reported that exosomal miR-155-5p can directly inhibit GATA binding protein 3 (GATA3) and tumor protein p53-induced nuclear protein 1 (TP53INP1) and can overcome paclitaxel resistance in gastric cancer cells [70]. 

In the future, it is expected that more sensitive and specific exosomal miRNA markers related to chemotherapy sensitivity will be identified. Additionally, the monitoring of miRNAs by liquid biopsy will enable the development of a method to evaluate the efficacy of chemotherapy more efficiently and select the appropriate timing of surgery on a real-time basis. However, clinical validation and standardization of the procedure are needed before liquid biopsy can be widely used on a routine basis. Early experiments analyzing liquid biopsies appear to be very promising in patients with advanced gastric cancer, but more prospective studies are needed to validate the efficacy of liquid biopsy and understand the molecular mechanisms underlying chemotherapy resistance.

## 9. Molecular Classification According to NGS Analysis

Over the past decade, next-generation sequencing (NGS) technology has been a powerful tool for studying the complexity of gastric cancers, with important implications for both the molecular characterization of the neoplasm and the therapeutic management of gastric cancer patients [71]. Indeed, the increasingly frequent integration of NGS in the molecular assessment of biological samples has the potential to greatly improve the selection of patients to be included in clinical trials of molecularly targeted drugs.

In fact, several amplicon-based NGS assays have been clinically approved and are currently being used to detect the most frequent and actionable genomic alterations in tumor samples [72]. The main NGS-based approaches (i.e., whole-genome sequencing [WGS], whole-exome sequencing [WES], RNA sequencing [RNA Seq], and targeted sequencing) have been systematically applied to characterize molecular alterations in gastric cancer [71]. This large amount of genomic, transcriptomic, and epigenomic data will significantly improve our understanding of the molecular landscape of gastric cancer, unravel its molecular heterogeneity, and pave the way for a comprehensive molecular classification of this complex disease, which will contribute to the development of new molecularly targeted drugs and the selection of patients to be included in clinical trials [73,74]. 

While the well-known Lauren classification criteria for gastric cancer were developed six decades ago according to histologic features (intestinal or diffuse), the use of genomic data has recently led to the development of new molecular classification schemes. In 2014, The Cancer Genome Atlas (TCGA) network developed a new molecular classification scheme using somatic cell copy number analysis, WES, DNA methylation profiling, messenger RNAseq, microRNA sequencing, and reverse-phase protein array profiling to characterize 295 localized and untreated gastric cancers [53]. According to TCGA results, gastric cancer can be classified into four molecular subtypes: (i) EBV-positive tumors (9%), (ii) MSI tumors (22%), (iii) genome stable (GS) tumors (20%), and (iv) tumors with chromosomal instability (CIN) (50%).

More importantly, this subclassification system was shown to have the potential to guide targeted therapy for different types of patients with gastric cancer subtypes. However, clinical data obtained by the Prodige group in France and the AIO group in Germany were negative for response discrimination regarding patients with diffuse (GS) versus intestinal (CIN) types [11,75,76]. No other robust predictive markers have been found for the GS and CIN groups. 

In 2015, the Asian Cancer Research Group (ACRG) proposed another molecular classification based on the evaluation of 300 gastric cancer samples in Korea by gene expression profiling, genome-wide copy number microarray, and targeted gene sequencing [77]. Four molecular subtypes of gastric cancer with different clinical and genomic features have been identified: (i) MSI tumors (23%); (ii) MSS with epithelial–mesenchymal transition features (MSS/EMT) tumors (15%); (iii) MSS with TP53 active (MSS/TP53þ) tumors (26%); and (iv) MSS with TP53 inactive (MSS/ P53-) tumors (36%). It should be noted that the differences between the TCGA and ACRG classifications are not perfect and uniform as they reflect differences in the approaches and platforms used and the ethnicity of the samples (i.e., global for TCGA and Korean for ACTG). However, it reveals molecular characteristics of gastric cancer that are not available from conventional histology-based classification, and thus is expected to improve our understanding of gastric cancer, improve treatment outcomes for gastric cancer patients, and potentially pave the way for better gastric cancer diagnosis and new drug development. A better understanding of the genomics of gastric cancer will allow optimization of treatment before or during NAC for individual patients. In other words, if this platform is incorporated into clinical research in the future and ultimately applied to routine medical practice, tailored NAC treatment for each patient will be possible.

## 10. Future Perspectives

Good predictive markers are expected to eliminate unnecessary and potentially detrimental NAC. In other words, more accurate predictive markers can identify patients at higher risk of locoregional recurrence or distant metastasis and increase the chances of curing the disease with lower toxicity by targeting key molecules and pathways. On the other hand, patients with biomarkers that suggest a low risk of local recurrence or distant metastasis, or markers of resistance to NAC, can proceed to surgery without receiving NAC. 

Tissue heterogeneity, a hallmark of gastric cancer, has been an obstacle to the development of predictive and prognostic biomarkers. Recently, Murugaesu et al. performed WES on eight patients, including more than 40 tumor regions, before and after neoadjuvant chemotherapy to assess the proportion of subclonal alterations in different tumor sites and to evaluate the degree of intratumor heterogeneity (ITH) in cancer [78]. Interestingly, more than half of all mutations were heterogeneously present in different tumor subclones. Using this approach, they reported that there is a strong correlation between the ITH index (mean of the proportion of heterogeneous mutations relative to the total number of mutations) and response to chemotherapy. This is consistent with the expectation that, from a biological perspective, tumors with high genomic heterogeneity would respond poorly to neoadjuvant therapy. Future larger, well-designed, prospective studies are needed to confirm the utility of ITH as a predictive and prognostic biomarker. From a diagnostic and monitoring perspective, the subclonal heterogeneity of gastric cancer suggests the utility of liquid biopsy, which can better reflect the subclonal mutational status in individual patients [79].

Furthermore, the heterogeneity of gastric cancer has necessitated different types of therapy. With advances in genomic and epigenomic research, further subclassification of gastric cancer into new molecular entities is expected to facilitate therapeutic decision making. In the foreseeable future, the integration of well-established clinicopathologic markers with modern molecular profiling will enable accurate prediction of NAC chemosensitivity in gastric cancer.

## 11. Conclusions

In this review, we provided an overview of promising biomarkers that could play a vital role in predicting the response to NAC in patients with LAGC. To date, various predictive factors for the therapeutic effect of conventional cytotoxic chemotherapy on gastric cancer have been identified. Some molecular targeted therapies and immune checkpoint inhibitors are now being introduced for gastric cancer, and their molecular markers such as human epidermal growth factor receptor 2 (HER2) and MSI are good predictive markers. However, these predictive factors have not yet been implemented in clinical practice for predicting the response to NAC.

More recently, gene-panel tests using NGS techniques have also been introduced into clinical practice. This new trend of personalized cancer treatment is expected to make the best use of the advantages of NAC by enabling the selection of refractory cases and effective prediction of treatment efficacy. Subsequently, a better understanding of the molecular characterization of gastric cancers will likely help employ targeted and biological therapies in relation to surgery, chemotherapy, and radiotherapy in gastric cancers, to improve outcomes in patient subsets with historically poor prognoses. In addition, further development of predictive markers will help define subgroups of patients who will benefit optimally from adjuvant treatment. Furthermore, the introduction of liquid biopsy methods will enable minimally invasive and reproducible preoperative sampling, which will lead to the appropriate selection and modification of treatment regimens based on prognostic risk and treatment-resistant biomarkers, thereby allowing NAC to become a more valid treatment strategy.

## Figures and Tables

**Table 1 biomedicines-10-01614-t001:** List of potential benefits and risks of neoadjuvant chemotherapy.

Possible Advantages	Possible Disadvantages
Downsizing or downstaging of the primary tumor	Delayed definitive surgery
Improvement of the possibility of subsequent R0 resection	Worsening general performance status
Eliminating systemic micrometastases	Chemotherapy-related peritumoral fibrotic reaction
Evaluation of a chemosensitivity-guide for adjuvant chemotherapy	Perioperative complication
More efficient delivery of chemotherapy due to prior surgical disruption of the vasculature	Disease progression (leads to inoperable disease)
Better tolerability than postoperative chemotherapy	

**Table 2 biomedicines-10-01614-t002:** Predictors of response to preoperative chemotherapy for advanced gastric cancer.

Biomarker	Chemotherapy	Samples	Cases	Method	Results	Author
DPD, TP, GADD45A	5-FU/cisplatin	Biopsy	61	Real-time PCR	High DPD levels were found more frequently in non-responding patients and were associated with worse survival. The combination of GADD45A and TP revealed the strongest predictive effect.	Napieralski et al. [17]
TS, MTHFR	5-FU	Blood	238	PCR	A significant survival benefit for the patients with NAC was found for the 2rpt/2rpt and 2rpt/3rpt genotypes	Ott et al. [18]
ERCC1	5-FU/cisplatin	Biopsy	38	PCR	ERCC1 mRNA levels had a statistically significant association with survival	Metzger et al. [19]
ERCC1	Platinum-based chemotherapy	Tissue	142	Immunohistochemistry	ERCC1 expression correlated with lack of histopathological response to NAC and was associated with OS	Fareed et al. [20]
DDB2/ERCC1	Docetaxel, cisplatin, S-1	Biopsy	43	Immunohistochemistry	DDB2- and/or ERCC1-high phenotype was significantly correlated with non-responding patients	Hirakawa et al. [21]
BAK	Docetaxel, cisplatin, S-1	Biopsy	69	Immunohistochemistry	BAK expression was predictive of chemotherapeutic responses and survival.	Kubo et al. [22]
MLH1	Fluorouracil-based doublet or triplet chemotherapy	Tissue	285	Immunohistochemistry	Loss of MLH1 was associated with chemoresistance and did not prolong survival following neoadjuvant chemotherapy.	Hashimoto et al. [23]
MSI	Platinum-based chemotherapy	Tissue	101	Immunohistochemistry	MSI-H phenotype was a favorable prognostic marker in patients with gastric cancer receiving NAC	Haag et al. [24]
MicroRNA (let-7i)	Folinic acid, fluorouracil, and oxaliplatin	Tissue	68	Quantitative RT-PCR.	Low let-7i expression was an unfavorable prognostic factor of OS.	Liu et al. [25]

Foot note: DPD, dihydropyrimidine dehydrogenase; TP, thymidine phosphorylase; TS, thymidylate synthase; MTHFR, 5,10-methylene-tetrahydrofolate reductase; ERCC1, gene excision repair cross-complementing; DDB2, damage DNA binding protein complex subunit 2; BAK, Bcl-2 homologous antagonist killer; MLH1, MutL homolog 1; MSI, microsatellite instability; 5-FU, 5-fluorouracil; RT-PCR, reverse transcription polymerase chain reaction; OS, overall survival; NAC, neoadjuvant chemotherapy.

## Data Availability

Not applicable.

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
