# Peer review of "Treatment Response Predictors of Neoadjuvant Therapy for Locally Advanced Gastric Cancer: Current Status and Future Perspectives"

_biomedicines, 2022, doi:10.3390/biomedicines10071614_

Round 1

Reviewer 1 Report

Dear authors, 

I've been carefully through your manuscript. It is indeed excellent..

To further strengthen your work I would strongly advice to create your own GRAPHICAL ABSTRACTS  mentioning the pathways of those biomarkers involved in NAC.

Best wishes

Author Response

Review #1

Comment: To further strengthen your work I would strongly advice to create your own GRAPHICAL ABSTRACTS  mentioning the pathways of those biomarkers involved in NAC.

Response: Thank you for your suggestion. I made the graphical abstract which could be a single, concise, pictorial and visual summary of biomarkers involved in NAC.

Reviewer 2 Report

In this brief and concise review, the authors present the latest knowledge on predictors of response to neoadjuvant therapy for gastric cancer. The authors present the advantages of neoadjuvant therapy in locally advanced gastric cancer as an effective treatment option that improves cure rates by reducing tumor size and preventing micro metastasis and recurrence. On the other hand, the authors point out some disadvantages of this therapy in patients refractory to neoadjuvant chemotherapy. The list of advantages and disadvantages is presented in a clear table format. Further, the authors summarize and discuss the progress in research on biomarkers of response to neoadjuvant chemotherapy in gastric cancer. The types of chemotherapy discussed and the corresponding biomarkers are also given in clear tabular format. The authors provide a future outlook and overview of promising biomarkers that may play a key role in predicting response to neoadjuvant chemotherapy in locally advanced gastric cancer. The authors also foresee a growing role for panel testing using next-generation sequencing technology, which will be increasingly introduced into clinical practice. This review is very informative and provides a glimpse into the future of further preoperative predictive markers for gastric cancer and modification of treatment regimens based on prognostic risk and biomarkers of treatment resistance for neoadjuvant chemotherapy.

Minor comments:

1.       Not every abbreviation is explained at the first appearance in the text. (line 51 CDDP, line 53 OS, line 57 MAGIC

2.       Lines 99-104 – is this a footnote to the table 2?

Author Response

Review #2

Comment: Not every abbreviation is explained at the first appearance in the text. (line 51 CDDP, line 53 OS, line 57 MAGIC

Response: We are sorry for not describing every abbreviation. I explained every abbreviation in the revised manuscript as follows,

“the Japan Clinical Oncology Group(JCOG), cisplatin (CDDP), overall survival (OS), Medical Research Council Adjuvant Gastric Infusional Chemotherapy (MAGIC), the Arbeitsgemeinschaft Internistische Onkologie (AIO) ,folinic acid, fluorouracil, and oxaliplatin (FOLFOX), the Capecitabine and Oxaliplatin Adjuvant Study in Stomach Cancer (CLASSIC), the Adjuvant Chemoradiotherapy in Stomach Tumors (ARTIST), and the Intergroup Trial in Adjuvant Chemotherapy for Adenocarcinoma of the Stomach (ITACA-S) trials, the ChemoRadiotherapy after Induction chemoTherapy In Cancer of the Stomach (CRITICS), human epidermal growth factor receptor 2 (HER2)”

Comment: Lines 99-104 – is this a footnote to the table 2?

Response: Yes, it is. For clarity, I added the word “footnote” just below the table in the revised manuscript.

This manuscript is a resubmission of an earlier submission. The following is a list of the peer review reports and author responses from that submission.